# Sleep Quality and Psychological Status in a Group of Italian Prisoners

**DOI:** 10.3390/ijerph17124224

**Published:** 2020-06-13

**Authors:** Giulia D’Aurizio, Angelica Caldarola, Marianna Ninniri, Marialucia Avvantaggiato, Giuseppe Curcio

**Affiliations:** 1Department of Biotechnological and Applied Clinical Sciences, University of L’Aquila, 67100 L’Aquila, Italy; giulia.daurizio@graduate.univaq.it (G.D.); angelicacaldarola91@gmail.com (A.C.); 2Penitentiary Institution-Casa Circondariale Lanciano, 66034 Chieti, Italy; ninnirimarianna@gmail.com (M.N.); marialucia.avantaggiato@giustizia.it (M.A.)

**Keywords:** depression, anxiety, insomnia, inmates, well-being

## Abstract

Prison could be considered a prolonged stressful situation that can trigger not only a dysregulation of sleep patterns but can also bring out psychiatric illness, such as anxiety and depression symptoms. Our study is aimed at exploring sleep quality and sleep habits in an Italian prison ward with three different security levels, and to attempt to clarify how anxiety state and the total time spent in prison can moderate insomnia complaints. There were 129 participants divided into three groups who enrolled in this study: 50 were in the medium-security prison ward (Group 1), 58 were in the high-security prison ward (Group 2) and 21 were in the medium-security following a protocol of detention with reduced custodial measures (Group 3). All participants filled in a set of questionnaires that included the Beck Depression Inventory (BDI-2), the State-Trait Anxiety Inventory (STAI), the Pittsburgh Sleep Quality Index (PSQI), and the Insomnia Severity Index (ISI). Based on their responses, we observed that all participants showed poor sleep quality and insomnia, mild to moderate depressive symptoms that tended to a higher severity in Groups 1 and 3, and the presence of clinically significant anxiety symptoms, mainly in Groups 1 and 3. Our study shows that increased anxiety state-level and the presence of mood alteration corresponds to an increase in both poor sleep quality and, more specifically, insomnia complaints. Finally, we propose that TiP (total time in prison) could have an interesting and stabilizing paradox-function on anxiety state and insomnia.

## 1. Introduction

Sleep is an adaptive, reversible, physiological, and behavioral process that plays an active and pivotal role in cognitive functioning and in behavior regulation. Lack of sleep or poor sleep quality can negatively affect several brain functions, such as executive function [1,2,3,4], emotional processing [5,6], and learning and memory consolidation [7,8,9]. Moreover, numerous studies have reported that prolonged sleep deprivation, as well as a more generally impaired sleep quality and quantity, correlates with some psychiatric diseases, such as depressed mood [10,11], anger and aggressive behavior [12], and anxiety [13,14]; while recovery from sleep loss directly reduces brain electroencephalographic arousal, improving sleep quality and quantity [15]. To date, most research has focused primarily on the relationship between sleep and cognitive functioning [16,17,18,19], psychiatric symptoms [20,21], and psychophysical well-being [22,23,24] in healthy populations or specific experimental groups, but few studies have probed this complex and potential bidirectional relationship in the prison population [25,26].

In these studies, it has been reported that in a particular environmental context, as the prisoner changes in sleep habits [27], potential sleep disturbances (i.e., insomnia) can progressively increase with time [28], but little is known about “which” potential factors could qualitatively and quantitatively modulate sleep and “how” psychological and psychiatric variables could impact sleep.

Typically, sleep disturbances that consequent to sleep pattern alterations represent a behavioral response linked to the prolonged exposure to extreme, stressful situations [29], such as a car crash [30], combat [31], or an earthquake [32,33]. In these extreme situations, anxiety also plays a pivotal role in the development of both sleep and behavior alterations [13,34]. According to this point of view, living in prison could be considered as a prolonged stressful situation that can trigger not only a dysregulation of sleep patterns, but can also bring out psychiatric symptoms, such as anxiety and/or depression. A recent review was conducted on a total sample of 22.790 detainees and documented the high rates of psychiatric morbidity among prisoners [35], showing the overall prevalence of 11.4% for major depression, 3.7% for psychotic illness, and 65% for personality disorder, while overall 12% female inmates were diagnosed with major depression, 4% with psychotic disorders, and 42% with personality disorders. In Italy, a cross-sectional study [36] of male prisoners reported a prevalence rate of 1.3% for psychosis, 2.4% for anxiety disorder, 5.4% for mood disorder, 4.1% for personality disorder, 2.6% for adjustment disorder, and 0.3% for mental retardation. In this vein, it would be necessary to elucidate whether psychiatric disorders are a pre-existing clinical profile in this population and then to represent potential causes of imprisonment. In a recent study [37], it was reported that nearly half of the prisoners interviewed seven days after their admission to the prison, reported substance use disorder and half of these had a morbidity for mental disorder (e.g., anxiety disorder). Moreover, between 4% and 6% of the prisoners did receive a diagnosis of psychiatric disorder in the year prior to the incarceration [38].

Overall, recent literature has widely demonstrated the association between sleep disturbances and anxiety, underlining how this relationship may be interpretable in two different ways [39], particularly when considering state anxiety. On one hand, anxiety disorders affect sleep process, disrupting physiological patterns and generating sleep loss [40]; on the other hand, they would seem to find the potential “anxiogenic effect” of sleep deprivation [41].

Moreover, several studies showed that, during imprisonment, inmates are much more likely to develop sleep disturbances in general, or more specifically, insomnia [42,43]. In this context, the high prevalence of depressive symptoms in prisoners [25,44], in addition to both a high level of anxiety and the prison’s environment, could explain the poor sleep quality and the insomnia symptoms.

In light of this, our study firstly aims at investigating sleep habits and the potential presence of psychopathological symptoms in a sample of Italian prisoners assigned to three experimental groups. Secondly, we aimed at elucidating the potential relationship between depression, anxiety, total time spent in prison (TiP) and the presence of insomnia through a regression analysis.

## 2. Materials and Methods

### 2.1. Participants

After obtaining the needed authorization by local penitentiary administration, in collaboration with the internal psychological team of the Penitentiary Institution of Lanciano (Chieti), we identified potential experimental subjects asking them to participate in the study. Based on the questionnaires and clinical interviews, we excluded participants that were not free of medication, with a known neurological condition and medical condition at the moment of the assessment, and with a known history of psychiatric disorder. Out of the 150 selected, 129 male prisoners (mean age 42.02 ± SD 11.7) accepted to participate and were enrolled. Participants were divided into three groups based on their conviction and on related prison security level in which they were living in: Group 1 was composed by inmates from the medium-security prison ward (*n* = 50; mean age 38.3 ± SD 10.73), Group 2 was composed by inmates from the high-security prison ward (*n* = 58; mean age 45.4 ± SD 11), and Group 3 was composed by inmates from medium-security prison ward living under detention with local mobility (*n* = 21; mean age 41.52 ± SD 13.64). We also controlled for the length of punishment, measuring for each participant the time spent in prison (TiP). All the participants lived in double cells, followed a similar daily routine with lights on at 7:00 a.m. and lights off at 10:00 p.m. (managed by prison guards) and with the same opportunity to have one hour of daylight exposure independently by security level. Some differences were present between groups with respect to personal freedom: Group 1 had the opportunity to momentarily leave their cell and move around the floor (if authorized by prison guards), while inmates of Group 2 did not and passed a great part of their life within the cell with closed door. Both groups’ prisoners were employed by social work within the jail with working hours planned by the Penitentiary Direction. Conversely, Group 3 had the opportunity to move around within the building inside which they live, having the chance to self-organize their life by deciding the preferred timing for each activity (cooking, eating, studying, having a shower, watching TV, etc.).

All participants were native Italian speakers with an education ranging between 5 and 8 years. This level of education assured that they were all able to read and understand the Italian version of the used questionnaires. Each inmate voluntarily accepted to participate in the study, no incentives to participate were allowed and they were explicitly informed to feel free to leave the study at any time without any kind of repercussion. All other prisoners and prison staff knew who participated in the study.

The whole investigation was approved by the Regional Institutional Review Board (Provveditorato Regionale Amministrazione Penitenziaria LAM, n. PE 0026156, 24 December 2018, released to G.C.) and was conducted according to the principles established in the Declaration of Helsinki.

### 2.2. Materials

All participants were asked to fill in a set of questionnaires in order to assess a psychological and behavioral profile together with sleep quality and quantity characteristics. The questionnaire-based assessment included the following scales.
Beck Depression Inventory II Edition (BDI-2) [45,46] is a 21-item multiple-choice self-report inventory for assessing affective-somatic (AS) and cognitive dimensions (C) of depression severity. The total score ranges from 0 to 63; in particular, a score from 10 to 18 indicates a mild to moderate depression, a score from 19 to 29 indicates a moderate to severe depression, and a score higher than 30 is indicative of a severe depression level [47].State-Trait Anxiety Inventory (STAI Form Y1 and Y2) [48,49] is a 40-item multiple-choice self-report inventory divided into two sub-tests, each having 20 items (for both state- and trait-anxiety) aimed at assessing and quantifying anxiety disorder in adults. All items are rated on a 4-point Likert scale. The recommended clinical cut-off is >46.Pittsburgh Sleep Quality Index (PSQI) [50,51] is a self-rated questionnaire for assessing the overall sleep quality for the past month. It provides info about seven different components (subjective sleep quality, C1; sleep latency, C2; sleep duration, C3; habitual sleep efficiency, C4; sleep disturbances, C5; use of sleep medications, C6; daytime dysfunction, C7) that when taken together form a global score. Usually, a PSQI global score ≥ 5 is an indicator of clinically significant sleep pathological alteration in at least two components or of moderate difficulties in more than three components.Insomnia Severity Index (ISI) [52,53] is a seven-item questionnaire widely used to quantify insomnia by evaluating some sleep aspects (i.e., sleep onset, sleep maintenance, early morning awakenings, sleep dissatisfaction, interference of sleep alterations with daytime functioning, sleep problems reported by others, level of distress secondary by the sleep alterations), referred to the past two weeks. The ISI total score ranges between 0 to 28, with a higher score indicating greater insomnia severity. More specifically, scores between 0 and 7 indicate non-significant insomnia, scores between 8 and 14 indicate subthreshold insomnia, scores between 15 and 21 indicate insomnia of moderate severity, and scores higher than 22 indicate severe insomnia [54,55].

### 2.3. Procedure

After enrolling in the study, each participant, under the supervision of a skilled experimenter (A.C.), filled in BDI-2, STAI Y1 and Y2, PSQI and ISI scales.

### 2.4. Data Analysis

#### 2.4.1. Kruskal-Wallis H test

A nonparametric Kruskal-Wallis H test was performed to test the differences between the three groups (Group 1, Group 2, Group 3) with respect to age and TiP (in years).

Moreover, in order to assess the differences across groups (Group 1, Group 2, Group 3) on psychological and sleep measures (Age and TiP), a nonparametric Kruskal-Wallis H test was run for total scores assessed by STAI Y1 and Y2, BDI-2, PSQI and ISI scales.

For both nonparametric analyses, alpha level was fixed to ≤0.05 and in case of significant main effects, the post-hoc Mann-Whitney test was carried out. A Bonferroni correction was also applied and thus, all effects were reported at 0.016 level of significance.

#### 2.4.2. Correlation Analysis

In order to determine the relationships between different scores at BDI-2, STAI Y1 and Y2, PSQI, ISI and TiP, a Spearman’s rank-order (r_s_) correlation was run. Alpha level was fixed to ≤0.05.

#### 2.4.3. Regression Analysis

Finally, based on Correlation analysis results, to further evaluate the effects of both psychopathological, and TiP on ISI Total Score, a Multiple Linear Regression analysis was applied, using BDI-2, STAI Y1 and Y2, and TiP as predictors. Alpha level was fixed to ≤0.05. All statistical analyses were performed using IBM SPSS Statistics for Macintosh, version 25.0 (IBM Corp., Armonk, NY, USA).

## 3. Results

### 3.1. Kruskal-Wallis H test

Age. Kruskal-Wallis H test showed a statistically significant difference regarding age between the three groups (H _(2)_ = 9.8; *p* = 0.007), with a mean rank age score of 53.6 for Group 1, 76 for Group 2, and 61.86 for Group 3. The pairwise Mann-Whitney test between groups showed a significant difference limited to the Group 1–Group 2 comparison (U = 955; Z = −3.3; *p* = 0.001).

Time in prison (TiP). Kruskal-Wallis H test showed a statistically significant difference with respect to TiP between the three groups (H _(2)_ = 7.74; *p* = 0.021), with a mean rank TiP score of 62 for Group 1, 71.2 for Group 2, and 45.3 for Group 3. The post-hoc Mann-Whitney test between groups showed a significant difference limited to the Group 2–Group 3 comparison (U = 354; Z = −2.86; *p* = 0.004).

Results of age and TiP are reported in Table 1.

STAI. Kruskal-Wallis H test showed a statistically significant difference in STAI Y1 between the three groups (H _(2)_ = 9.6; *p* = 0.008), with a mean rank STAI Y1 score of 72.9 for Group 1, 53.83 for Group 2, and 77.07 for Group 3. The pairwise Mann-Whitney test between groups showed a significant difference between Group 1 and Group 2 (U = 1033; Z = −2.57; *p* = 0.01), and between Group 2 and Group 3 (U = 378; Z = −2.56; *p* = 0.01).

Conversely, there was no statistically significant difference in the Y2 version of the questionnaire (H _(2)_ = 4.46; *p* = 0.1).

BDI-2. Kruskal-Wallis H test showed a statistically significant difference in BDI-2 between the three groups (H _(2)_ = 9.98; *p* = 0.007), with a mean rank STAI Y1 score of 72.76 for Group 1, 53.66 for Group 2, and 77.83 for Group 3. The post-hoc Mann-Whitney test showed a significant difference between Group 1 and Group 2 (U = 1022.5; Z = −2.64; *p* = 0.008), and between Group 2 and Group 3 (U = 379; Z = −2.56; *p* = 0.011).

PSQI. Kruskal-Wallis H test showed a statistically significant difference in PSQI total score between the three groups (H _(2)_ = 10.15; *p* = 0.006), with a mean rank STAI Y1 score of 76.42 for Group 1, 53.82 for Group 2, and 68.7 for Group 3. The post-hoc Mann-Whitney test showed a significant difference limited to the Group 1–Group 2 (U = 951.5; Z = −3.08; *p* = 0.002) comparison.

ISI. There was no statistically significant difference between the insomnia complaints groups (H _(2)_ = 5.62; *p* = 0.06)

Results of psychological and sleep questionnaires are reported in Table 2.

### 3.2. Correlation Analysis

Spearman’s rank-order correlation analysis showed a significant negative correlation between TiP and STAI Y1 (r_s_ = −0.2; *p* = 0.042), between TiP and PSQI (r_s_ = –0.2; *p* = 0.03) and between TiP and ISI total score (r_s_ = −0.24; *p* = 0.006). With respect to STAI, we found a significant positive correlation between STAI Y1 scale and BDI-2 (r_s_ = 0.56; *p* < 0.0001), PSQI (r_s_ = 0.53; *p* < 0.0001), and ISI (r_s_ = 0.54; *p* < 0.0001) total score. A positive correlation was also found between Y2 scale and BDI-2 (r_s_ = 0.63; *p* < 0.0001), PSQI (r_s_ = 0.5; *p* < 0.0001), and ISI (r_s_ = 0.5; *p* < 0.0001) total score. Moreover, correlation analysis showed a significant positive correlation between BDI-2 and both PSQI (r_s_ = 0.45; *p* < 0.0001), and ISI total score (r_s_ = 0.41; *p* < 0.0001).

Instead, as expected, the statistical analysis indicated a significant positive correlation between PSQI and ISI (r_s_ = 0.6; *p* < 0.0001).

Results are reported in Table 3.

### 3.3. Regression Analysis

Multiple linear regression analysis results showed that the model explains 31% of the variance (R^2^ = 0.31, F_(4,125)_ = 13.6, *p* < 0.0001) of ISI total score. It was found that both STAI Y1 and BDI-2 total score significantly and positively predicted ISI total score (*β* = 0.3, t = 2,52, *p* = 0.013; β = 0.25, t = 2,47, *p* = 0.015, respectively), indicating that higher levels of state anxiety and higher depression scores predict an impoverished sleep quality. Conversely, neither TiP (*β* = −0.11, t = −1.45, *p* = 0.14) nor STAI Y2 (*β* = 0.55, t = 0.44, *p* = 0.66) appeared to act as predictors of ISI total score.

## 4. Discussion

The first aim of this study was to focus on sleep quality and potential psychopathological symptoms in the complex and multifaceted environment of prison, where particular real-life conditions and newly imposed rules suddenly change inmates’ habits.

With respect to depressive symptoms, results show that all experimental subjects report a clinical total score from mild to moderate, with symptoms that tend to be of higher severity both in Groups 3 and 1 than Group 2, respectively.

Furthermore, regarding anxiety disease, our results reveal the presence of clinically significant symptoms, in both state and trait components, with particular involvement of both Groups 1 and 3.

Therefore, in the light of the above, we can derive the first conclusion: generally, our experimental sample is characterized by both depressive and anxiety symptoms and these clinical features are more present in experimental Groups 1 and 3, whose members spend their jail time for minor crimes in the medium-security prison ward, but not in the prisoners’ group that spend their time in the high-security ward. It is well known that many factors must be taken into consideration when dealing with prisoners and mental health (such as international differences, prison setting, demographics, etc.) and, at the same time, the early phase of imprisonment is a vulnerable period with a moderately higher incidence of adjustment disorders [56]. On the contrary, it has also been reported that inmates sentenced to an indefinite term of imprisonment show a relevant increase in mood disorders and substance abuse [57]. Present data can be explained in different ways. As a first, we could hypothesize that the complex environmental context, in which these people live could positively modulate their ability to adapt to the new context of prison life. Alternatively, their mood stability could be explained on the basis of a different sense of resignation to the inevitability of their future life within the jail. This second hypothesis seems to be supported by previous studies linking personality traits, acceptance-resignation and emotional discharge, coping style, and tendency to psychopathological symptomatology [58]. This interesting aspect must be further investigated in future studies, in which personality and behavioral characteristics should also be investigated.

With respect to sleep quality, according to a previous study [43,59], our results demonstrate and confirm the poor sleep quality in the prison population and the presence of insomnia symptoms, despite being clinically subthreshold.

Importantly, poor sleep quality would seem to follow the same trend that emerged for the psychopathological symptoms: on average, the groups in the medium-security prison ward report worse sleep quality than groups in the high-security ward.

The bidirectional relationship between anxiety and sleep disorders are widely described, but it continues to be unclear which one of the two factors dysregulates the other, and if in the prison context another variable can exist, such as the TiP, capable of negatively affecting sleep patterns, and anxiety state. Our analysis results would show that TiP could have a stabilizing paradox-function on anxiety state and insomnia, considering that its increase would correspond to a decrease in both anxiety state symptoms and insomnia levels. Results also show that increased anxiety state-level and the presence of mood alteration corresponds to an increase in both poor sleep quality and, more specifically, insomnia complaints.

The importance of clarifying and studying sleep habits, and related-anxiety and depression disorders within the prison’s environment appears important in light of some fundamental aspects, such as aggressive behavior and a rate of suicidal behavior that continues to be high among prisoners [60]. It is plausible to hypothesize that a reduction of anxiety disorders and a promotion of a correct sleep hygiene could positively modulate the reduction of altered emotional responses (i.e., aggressive behavior, suicidal behavior, and mood disorders). In this context, future research should identify adequate intervention strategies, which should aim to improve the inmates’ quality of life.

In our study, several limitations need to be acknowledged, in particular the small sample size, the gender bias (in our study sample all the participants were men), and the unequal number of subjects among the three experimental groups. Moreover, it should also be taken into consideration that, despite the small differences in the daily organization between security levels (sleep-wake rhythms, exposure to light or noise, etc.) undoubtedly prisoners of Group 3 (medium-security, living under detention with local mobility) were more “free” to move around, having the chance to self-organize their life by deciding the preferred timing for each activity (cooking, eating, studying, having a shower, watching TV, etc.). Nonetheless, we should also bear in mind that these limitations depend on the objective experimental difficulties encountered during the study and, in particular, are direct consequences of the severe rules regulating the prison environment.

## 5. Conclusions

In conclusion, based on the above described results, we observed that our study sample showed poor sleep quality and insomnia symptoms, mild to moderate depressive symptoms that tended to a higher severity in Groups 1 and 3, and the presence of clinically significant anxiety symptoms, mainly in Groups 1 and 3. Specifically, the results highlighted that increased anxiety state-level and the presence of mood alteration corresponds to an increase in both poor sleep quality and insomnia complaints.

The findings from the present study do propose that TiP could have an interesting and stabilizing paradox-function on state anxiety and insomnia.

## Figures and Tables

**Table 1 ijerph-17-04224-t001:** Mean, standard deviation and median of age and TiP (total time in prison) study sample.

Variables	Group 1 (*n* = 50)	Group 2 (*n* = 58)	Group 3 (*n* = 21)	*p*
Age	38.3 ± 10.73 (36)	45.41 ± 10.95 (45)	41.52 ± 13.64 (39)	0.007
TiP	4.16 ± 3.74 (3)	4.82 ± 3.77 (3)	2.5 ± 2.6 (2)	0.021

**Table 2 ijerph-17-04224-t002:** Mean, standard deviation and median psychopathological, and sleep questionnaires’ total scores of the study sample.

Variables	Group 1 (*n* = 50)	Group 2 (*n* = 58)	Group 3 (*n* = 21)	*p*
STAI Y1	49.4 ± 4.6 (46.5)	43.6 ± 12.6 (42.5)	49.45 ± 11.9 (53)	0.008
STAI Y2	48.04 ± 7.37 (48)	45.67 ± 8.1 (44)	47.5 ± 7.5 (47)	n.s.
BDI-2	15.24 ± 9.2 (14)	11.4 ± 8.9 (9)	15.6 ± 7.85 (16)	0.007
PSQI	11.16 ± 4.6 (12)	9 ± 3.4 (8.5)	10.23 ± 3 (10)	0.006
ISI	11.38 ± 4.6 (11.5)	9.13 ± 8 (8)	13.23 ± 8 (12)	n.s.

Note: STAI Y1 = State-Trait Anxiety Inventory-State subscale, STAI Y2 = State-Trait Anxiety Inventory-Trait subscale, BDI-2 = Beck Depression Inventory II Edition, PSQI = Pittsburgh Sleep Quality Index, ISI = Insomnia Severity Index.

**Table 3 ijerph-17-04224-t003:** Spearman’s r_s_ (and related level of significance) between TiP and STAI Y1 and Y2, PSQI, ISI, and BDI-2 total scores.

Variables	TiP		BDI-2		STAI Y1		STAI Y2		PSQI		ISI	
	r_s_	*p*	r_s_	*p*	r_s_	*p*	r_s_	*p*	r_s_	*p*	r_s_	*p*
TiP	1		−0.07	0.5	−0.2	**0.042**	−0.07	0.45	−0.2	**0.03**	−0.24	**0.006**
BDI-2	−0.07	0.5	1		0.56	**0.000**	0.63	**0.000**	0.45	**0.000**	0.41	**0.000**
STAI Y1	−0.2	**0.042**	0.56	**0.000**	1		0.72	**0.000**	0.53	**0.000**	0.54	**0.000**
STAI Y2	−0.07	0.45	0.63	**0.000**	0.72	**0.000**	1		0.5	**0.000**	0.5	**0.000**
PSQI	−0.2	0.03	0.45	**0.000**	0.53	**0.000**	0.5	**0.000**	1		0.6	**0.000**
ISI	−0.24	**0.006**	0.41	**0.000**	0.54	**0.000**	0.5	**0.000**	0.6	**0.000**	1	

Note: STAI Y1 = State-Trait Anxiety Inventory-State subscale, STAI Y2 = State-Trait Anxiety Inventory-Trait subscale, BDI-2 = Beck Depression Inventory II Edition, PSQI = Pittsburgh Sleep Quality Index, ISI = Insomnia Severity Index. Significant correlations are indicated in bold.

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
