# Peer review of "Sleep Quality and Psychological Status in a Group of Italian Prisoners"

_ijerph, 2020, doi:10.3390/ijerph17124224_

Round 1
Reviewer 1 Report
Thank you for the opportunity to review this manuscript- sleep deficiency in correctional institutions is a significant and understudied area – this paper will be a welcome addition.
I have several recommendations I encourage you to consider towards strengthen the paper yet further:
Line 43- one or two citations to more comprehensive reviews would be appropriate here. The current study of adolescents is insufficient to support this claim.
Lines 46-70: The general background could be strengthened by more discussion presenting the different organizational structures, policies, and environments (eg control over lighting, shared or single cells, etc) in these studies. Contextual factors can play a strong role in sleep and it needs to be stated that we cannot assume institutions across countries are the same. There is unfortunately a lack of information about context in other prison studies and this needs to be pointed out as a limitation to the existing evidence-base.
Lines 51-57: is there literature to give perspective on the incidence and prevalence of anxiety and depression on inmates both acquired in the institution and also as they enter into the system? Pre-existing psychiatric problems are a significant risk for incarceration – eg(https://www.ncbi.nlm.nih.gov/pubmed/31480926).
Line 75: please provide detail about how participants were approached. Were other prisoners and prison staff blind to who participated or did not? Was there an incentive to participate? These questions all speak to participant bias.
Line 77: there is grammatical error
Line 80: You state they were assigned to ‘experimental‘ groups but as there was no intervention perhaps this term in misleading. Would it be more accurate to state they were sorted by three categories related to the terms of their incarceration?
Materials: No data is reported about environmental features of prisoners’ daytime and sleep environment for each category. Environmental factors are critical in sleep physiology and it is important to know if they all had the same opportunity for daylight exposure, and nighttime dark and quiet. Also did they all have single cells or were some co-sleeping? All of these can be cofounders. If they were not in the same environment then it needs to be detailed as a study limitation.
Line 120- were all of the participants literate and able to read and understand the questionnaires or were they orally administered? Can you give more detail about whether the forms were translated for this population and tested for reliability of so? Or were they administer in the English version? If so how might completing a questionnaire in a second language have impacted the participants’ performance?
Line 193 onward: I think without discussion of the environmental context it would be premature to draw conclusions or speculate about the impact of time in prison. It may well be that those in high-security had single cells, or more structured daylight exposure, or darker nighttime environments etc compared to the other groups and that this was what make sleep more efficient for those participants.
This is an important study- too little is known about sleep in correctional institutions and that is a tragedy given that sleep is a modifiable risk factor for so many negative phenomena.
Author Response
Thank you for the opportunity to review this manuscript- sleep deficiency in correctional institutions is a significant and understudied area – this paper will be a welcome addition.
I have several recommendations I encourage you to consider towards strengthen the paper yet further:
Authors’ Answer (AA): We thank the Reviewer for his/her interest in our manuscript.
Line 43- one or two citations to more comprehensive reviews would be appropriate here. The current study of adolescents is insufficient to support this claim.
AA: We thank the Reviewer for this comment: it has been changed in the text.
Lines 46-70: The general background could be strengthened by more discussion presenting the different organizational structures, policies, and environments (eg control over lighting, shared or single cells, etc) in these studies. Contextual factors can play a strong role in sleep and it needs to be stated that we cannot assume institutions across countries are the same. There is unfortunately a lack of information about context in other prison studies and this needs to be pointed out as a limitation to the existing evidence-base.
AA: We thank the Reviewer for this comment: we changed the text accordingly, also to answer to other concerns (see following points raised by the same Reviewer), in order to clarify environmental differences and daily planning of activities. These aspects have also been acknowledged as possible limitations of the study.
Lines 51-57: is there literature to give perspective on the incidence and prevalence of anxiety and depression on inmates both acquired in the institution and also as they enter into the system? Pre-existing psychiatric problems are a significant risk for incarceration – eg(https://www.ncbi.nlm.nih.gov/pubmed/31480926).
AA: We thank the Reviewer for this comment. These aspects have been included in the manuscript together with some additional references.
Line 75: please provide detail about how participants were approached. Were other prisoners and prison staff blind to who participated or did not? Was there an incentive to participate? These questions all speak to participant bias.
AA: The lacking info have now been included in the manuscript in the Participants section.
Line 77: there is grammatical error
AA: The error has been fixed.
Line 80: You state they were assigned to ‘experimental‘ groups but as there was no intervention perhaps this term in misleading. Would it be more accurate to state they were sorted by three categories related to the terms of their incarceration?
AA: The misleading word “experimental” referred to the group has been deleted and some more info of the three categories of incarceration rules have been included. In this way we feel to have also answered to the following point about environmental features of prisoners’ life.
Materials: No data is reported about environmental features of prisoners’ daytime and sleep environment for each category. Environmental factors are critical in sleep physiology and it is important to know if they all had the same opportunity for daylight exposure, and nighttime dark and quiet. Also did they all have single cells or were some co-sleeping? All of these can be cofounders. If they were not in the same environment then it needs to be detailed as a study limitation.
AA: This point has been addressed together with the previous one. Some of these issues have been included in Discussion as possible limitations. We thank the Reviewer for these constructive criticisms.
Line 120- were all of the participants literate and able to read and understand the questionnaires or were they orally administered? Can you give more detail about whether the forms were translated for this population and tested for reliability of so? Or were they administer in the English version? If so how might completing a questionnaire in a second language have impacted the participants’ performance?
AA: All the participants were native Italian speaker and able to read and understand the Italian version of the questionnaires administered: this is now specified in the manuscript.
Moreover, they filled in all translated versions of the questionnaires (also this info has been added to the revised manuscript).
Finally, all the questionnaires administered were used in previous studies on prisoners and/or populations that experienced stressful situations (i.e. Elger B. S. (2003). Does insomnia in prison improve with time? Prospective study among remanded prisoners using the Pittsburgh Sleep Quality Index. Medicine, science, and the law, 43(4), 334–344. doi.org/10.1258/rsmmsl.43.4.334; Elger, B. S.; Sekera, E. Prospective evaluation of insomnia in prison using the Pittsburgh Sleep Quality Index: Which are the factors predicting insomnia? International Journal of Psychiatry in Clinical Practice 2009, 13(3), 206–217. doi:10.1080/13651500902812043; Pigeon, W. R., Campbell, C. E., Possemato, K., & Ouimette, P. (2013). Longitudinal relationships of insomnia, nightmares, and PTSD severity in recent combat veterans. Journal of psychosomatic research, 75(6), 546–550.doi.org/10.1016/j.jpsychores.2013.09.004) assuring a satisfactory level of reliability of such measurements in this context.
Line 193 onward: I think without discussion of the environmental context it would be premature to draw conclusions or speculate about the impact of time in prison. It may well be that those in high-security had single cells, or more structured daylight exposure, or darker nighttime environments etc compared to the other groups and that this was what make sleep more efficient for those participants.
AA: As said before with respect to similar concerns, these warnings have been included in the Discussion as potential limitations to the study.
This is an important study- too little is known about sleep in correctional institutions and that is a tragedy given that sleep is a modifiable risk factor for so many negative phenomena.
AA: We thank the Reviewer for his/her positive interest in our work.
Reviewer 2 Report
The topic of the paper is interesting but I have some concerns at this stage that do not make me inclined to endorse it.
Data analysis - we have no information about the distributional patterns of the variables scores, thus it is impossible to assess whether parametric statistics was the right choice (e.g., ANOVA) - i any assumptions for this test were violated, a non-parametric approach should have been sought.
The major shortcoming I see is that there seems to be no discussion whatsoever about confounding factors. For instance, there is no reference to possible effects of actual environmental factors and differences of context between Groups 1-2-3 on sleep quality. Is there any info about noise/light levels inmates are exposed to during nighttime in the different conditions? What about cell occupancy - could it be that lower-security group has more inmates in the same cell, while higher-security sleep alone? This factor should also be controlled for.
The full details of the ethical approval should be reported and also more information on the process to seek informed consent, considering the potentially vulnerable group of participants targeted.
Author Response
The topic of the paper is interesting but I have some concerns at this stage that do not make me inclined to endorse it.
Authors’ Answer (AA): We thank the Reviewer for his/her interest in our manuscript.
Data analysis - we have no information about the distributional patterns of the variables scores, thus it is impossible to assess whether parametric statistics was the right choice (e.g., ANOVA) - i any assumptions for this test were violated, a non-parametric approach should have been sought.
AA: The Reviewer is right. In reality, during manuscript preparing we launched both kind of analyses and (as the Reviewer will see by comparing “previous” and “current” results in the Track changes version of the Ms.) no difference became evident. Thus, we preferred the parametric approach in order to have the opportunity to run an ANCOVA considering Age and TiP as covariates. For completeness of information and fairness, it should be said that normality was ascertained only for some of the considered variables (i.e. Age, STAI-Y1) whilst some others appeared very close to the normality (i.e, STAI-Y2, PSQI)
Nonetheless, we understand the statistical criticism by Reviewer and accept his/her suggestions: thus, in this revised version of the manuscript non-parametric analyses have been included instead of ANOVA and ANCOVA, and the same was done for correlation (Spearman instead of Pearson).
Finally, according to Li et al., 2013 (Are Linear Regression Techniques Appropriate for Analysis When the Dependent (Outcome) Variable Is Not Normally Distributed?) and taking into account that: 1) most of our variables were normally distributed or very close to the normality (as said before), 2) basically nonparametric analyses confirmed the same results highlighted by parametric ones (rather highlighting one more) and 3) that literature is divided on this issue, Multiple Linear Regression analyses appeared to be the most correct analysis to be applied here and has been confirmed in this revised version of the manuscript.
The major shortcoming I see is that there seems to be no discussion whatsoever about confounding factors. For instance, there is no reference to possible effects of actual environmental factors and differences of context between Groups 1-2-3 on sleep quality. Is there any info about noise/light levels inmates are exposed to during nighttime in the different conditions? What about cell occupancy - could it be that lower-security group has more inmates in the same cell, while higher-security sleep alone? This factor should also be controlled for.
AA: We thank the Reviewer for these suggestions. Also to answer to similar observations raised by Reviewer #1 we included such info on Participants paragraph, and in the Discussion as possible confounding factors and limitations to the study.
The full details of the ethical approval should be reported and also more information on the process to seek informed consent, considering the potentially vulnerable group of participants targeted.
AA: Full details have been included.
Reviewer 3 Report
This is an interesting brief-report describing sleep quality and psychological status in a group of Italian prisoners assigned to three different security levels. The data were collected using a set of subjective questionnaires (PSQI, ISI, STAI, BDI-2) and several parametric tests were used to explore the effects on investigated variables.
Results confirm both (a) sleep pattern alteration in the context of an extreme and stressful situation and (b) the bidirectional relation between sleep and mental health. Furthermore, the novelty of this study is represented by the role of the specific type of stressful situation in the development and evolution of sleep and psychological problems.
The following minor points need to be addressed to improve the readability and interpretation of the study.
- Which is the exact number of participants? 129 or 126? Please make this information coherent within the text and in the tables of results
- Line 44: a few studied did probe > a few studied probed
- Please keep verb tenses consistent in the text
- Line 62: the sentence “the potential “anxiogenic effect” of sleep deprivation on anxiety” is redundant. Consider removing the expression “anxiogenic” or “on anxiety”
- Line 77: exclude we > we excluded
- Line 94,99,193,110: please write the citation number inside square brackets
- The author should also mention the gender bias in the limitation section at the end of the Discussion
- The authors have good knowledge of this scientific matter, as demonstrated by an adequate literature background and their speculative hypotheses. However, the novelty relative to the unexpected worse conditions in Group 1 and 3 (medium-security) is not adequately discussed. I consider that the discussion should better explain the original results based on the previous literature.
- Finally, I strongly suggest that the present manuscript should be checked by native English speaker before publication.
Author Response
This is an interesting brief-report describing sleep quality and psychological status in a group of Italian prisoners assigned to three different security levels. The data were collected using a set of subjective questionnaires (PSQI, ISI, STAI, BDI-2) and several parametric tests were used to explore the effects on investigated variables.
Results confirm both (a) sleep pattern alteration in the context of an extreme and stressful situation and (b) the bidirectional relation between sleep and mental health. Furthermore, the novelty of this study is represented by the role of the specific type of stressful situation in the development and evolution of sleep and psychological problems.
Authors’ Answer (AA): We thank the Reviewer for his/her interest and positive impression on our manuscript.
The following minor points need to be addressed to improve the readability and interpretation of the study.
Which is the exact number of participants? 129 or 126? Please make this information coherent within the text and in the tables of results
AA: We thank the Reviewer for his/her concern: it has been changed in the text.
Line 44: a few studied did probe > a few studied probed
AA: We thank the Reviewer for his/her concern: it has been changed in the text.
Please keep verb tenses consistent in the text
AA: We thank the Reviewer for his/her concern: the text has been changed accordingly.
Line 62: the sentence “the potential “anxiogenic effect” of sleep deprivation on anxiety” is redundant. Consider removing the expression “anxiogenic” or “on anxiety”
AA: We thank the Reviewer for his/her concern: it has been changed in the text.
Line 77: exclude we > we excluded
AA: We thank the Reviewer for his/her concern: it has been changed in the text.
Line 94,99,193,110: please write the citation number inside square brackets
AA: We thank the Reviewer for his/her concern: it has been changed in the text.
The author should also mention the gender bias in the limitation section at the end of the Discussion
AA: We thank the Reviewer for his/her suggestion: the issue has been included in the Discussion.
The authors have good knowledge of this scientific matter, as demonstrated by an adequate literature background and their speculative hypotheses. However, the novelty relative to the unexpected worse conditions in Group 1 and 3 (medium-security) is not adequately discussed. I consider that the discussion should better explain the original results based on the previous literature.
AA: Present results have now been discussed more in depth..
Finally, I strongly suggest that the present manuscript should be checked by native English speaker before publication.
AA: the whole manuscript has been checked by a native English speaker.
Round 2
Reviewer 2 Report
I think the authors have made sufficient improvement for the paper to be endorsed.